# LRRpredictor—A New LRR Motif Detection Method for Irregular Motifs of Plant NLR Proteins Using an Ensemble of Classifiers

**DOI:** 10.3390/genes11030286

**Published:** 2020-03-08

**Authors:** Eliza C. Martin, Octavina C. A. Sukarta, Laurentiu Spiridon, Laurentiu G. Grigore, Vlad Constantinescu, Robi Tacutu, Aska Goverse, Andrei-Jose Petrescu

**Affiliations:** 1Department of Bioinformatics and Structural Biochemistry, Institute of Biochemistry of the Romanian Academy, Splaiul Independentei 296, 060031 Bucharest, Romania; eliza.martin@biochim.ro (E.C.M.); laurentiu.spiridon@biochim.ro (L.S.); vlad.ion.constantinescu@gmail.com (V.C.); robi.tacutu@gmail.com (R.T.); 2Laboratory of Nematology, Wageningen University and Research, 6700ES Wageningen, The Netherlands; octavina.sukarta@wur.nl; 3Space Comp SRL, 041512 Bucharest, Romania; laur@itprod.eu

**Keywords:** leucine-rich repeat prediction, supervised learning, LRR motif, LRR structure, NOD-like receptors, R proteins

## Abstract

Leucine-rich-repeats (LRRs) belong to an archaic procaryal protein architecture that is widely involved in protein–protein interactions. In eukaryotes, LRR domains developed into key recognition modules in many innate immune receptor classes. Due to the high sequence variability imposed by recognition specificity, precise repeat delineation is often difficult especially in plant NOD-like Receptors (NLRs) notorious for showing far larger irregularities. To address this problem, we introduce here LRRpredictor, a method based on an ensemble of estimators designed to better identify LRR motifs in general but particularly adapted for handling more irregular LRR environments, thus allowing to compensate for the scarcity of structural data on NLR proteins. The extrapolation capacity tested on a set of annotated LRR domains from six immune receptor classes shows the ability of LRRpredictor to recover all previously defined specific motif consensuses and to extend the LRR motif coverage over annotated LRR domains. This analysis confirms the increased variability of LRR motifs in plant and vertebrate NLRs when compared to extracellular receptors, consistent with previous studies. Hence, LRRpredictor is able to provide novel insights into the diversification of LRR domains and a robust support for structure-informed analyses of LRRs in immune receptor functioning.

## 1. Introduction

The leucine-rich-repeat (LRR) domains are present in all of the tree of life branches. As they are involved in protein–protein interactions, LRR domains are found in receptors having a vast number of functions such as pathogen detection, immune response propagation, hormone perception, enzyme inhibition, or cell adhesion [1]. In both plants and mammals, a number of studies have detailed adverse effects associated with mutations in the LRR domains such as that reported for various immune-related receptors, resulting in compromised functions and enhanced disease progression [2]. For example, mutating a single residue in the LRR domain of the rice Pita receptor results in complete loss of recognition against the fungus *Magnaporthe grisea* [3] while mutations in the metazoan NLRC4-LRR contributes to autoinflammatory disease phenotypes [4]. Additionally, mutations in the LRRK2 kinase enzyme, lead to Parkinson’s disease and other associated inflammatory diseases [5,6], whereas mutations in leucine-rich proteoglycans have been previously shown to be involved in osteoarthritis [7], and last but not least PRELP mutations might have a role in Hutchinson–Gilford, an accelerated progeroid syndrome characterized by premature aging [8]. Hence, understanding the structural aspects of binding properties and specificities of LRR domains opens wide possibilities for receptor engineering with vast implications not only for improved crop resistance to plant diseases, but also for a wide range of medical applications.

In innate immunity, LRR modules are found in various domain organizations in many receptor classes such as plant receptor-like kinases (RLK), receptor-like proteins (RLP), NOD-like receptors (NLR), or metazoan NLR and Toll-like receptors (TLR). In plant basal immunity, LRR N-terminal domains face the extracellular environment and are found in either receptor-like kinases (RLK) or receptor-like proteins (RLPs) depending on the presence or absence of a C-terminal kinase domain on the cytosolic side of the receptor. By contrast, LRRs constitute the C-terminal domains of intracellular NOD-like receptors (NLR), also known as resistance (R) proteins, and face the cytosolic environment to mediate resistance against specific pathogens. Depending on their N-terminal domain, which is either a coiled-coil (CC) or a toll-like receptor domain (TIR), R proteins fall into two main NLR classes: the CNL and TNL receptors, respectively [9]. Both these classes contain however a central nucleotide binding domain (NBS) which acts as a ‘switch’ that changes its conformation upon ADP/ATP binding [9,10]. Metazoan NLRs show a similar organization with plant NLRs. They encode a variety of N-terminal ‘sensors’ (caspase activation and recruitment domains—CARD, baculovirus inhibitor of apoptosis repeat—BIR, etc.), the central ‘switch’ STAND domain (signal transduction ATPases with numerous domains) - NBS/NACHT domain (NAIP (neuronal apoptosis inhibitory protein), CIITA (MHC class II transcription activator), HET-E (incompatibility locus protein from Podospora anserina) and TP1 (telomerase-associated protein)) and the LRR domain at the C-terminal end. Last but not least, we mention here the metazoan toll-like receptors (TLRs) that have an extracellular LRR sensor domain as seen in the RLK/RLP case and a TIR domain on the cytosolic side involved in signal transduction [11].

From a structural point of view LRR domains have a solenoidal ‘horseshoe’ like 3D architecture composed of a variable number of repeats varying each from ≈15 to ≈30 amino acids in length. Repeats are held together through a network of hydrogen bonds which forms a beta sheet located on the ventral side of the ‘horseshoe’. This is generated by a conserved sequence pattern named the LRR motif that in its minimal form is of the type ‘LxxLxL’ where L is generally leucine and to a lesser degree other hydrophobic amino acids [12]. Comprehensive sequence analysis of LRR immune receptors resulted in several classifications of LRR domains showing preferred amino acid conservation outside the minimal motif such as the two type classification proposed by Matsushima et al. [13] for TLR receptors or the seven type classification proposed by Kobe and Kajava [14] for all known LRR domains across all Kingdoms. However, exceptions to such rules are frequent as revealed by the Hidden Markov Model approach carried out by Ng et al. [15]. This highlighted the fact that most of the analyzed classes of human proteins containing LRR domains also display many irregular motifs alongside repeats showing the well-defined class specific motif [15].

While the above mentioned receptor classes were shown to present LRR irregularities [15], studies on plant NLR proteins such as Lr10 and Pm3 from wheat, Rx1 and Gpa2 from potato, or ZAR1 from Arabidopsis show that their LRR domains have a far more variable and irregular structure than their extracellular counterparts [16,17,18,19,20,21,22]. These factors combined contribute to the challenge for the accurate prediction of LRR motifs in plant NLRs.

A proper annotation of each LRR motif in a given LRR domain is instrumental in generating an accurate 3D model [12,23] and by this in properly defining the domain surface and identifying potential protein–protein interaction interfaces. An illustrative example is the conservation mapping performed by Helft et al. in 2011, which was used to identify new interaction partners of plant RLPs and RLKs by studying conserved 3D relationships among amino acids inferred from annotation of LRR repeats [24].

Based on our previous work, identifying the individual true motifs in a LRR domain is hindered by the following: (a) in its minimal form, a ‘LxxLxL’ pattern is trivial and frequently occurs randomly in any protein; (b) in many cases several ‘LxxLxL’ patterns do overlap in less than 15 aa range in NLR-LRRs making the precise delineation difficult; (c) the number of 3D experimental structures from which to learn is low; and (d) this small 3D learning set is class and phyla biased—as around half of the structures are of mammalian origin while plant NLRs only have one recently documented structure [21,22].

Thus, given the above described indeterminacies the precise LRR motif identification becomes the most problematic step in the correct repeat delineation within a LRR domain. This also explains why LRR domains and their individual repeats are poorly annotated in genomes or protein databases in contrast to the better annotated, relatively more conserved NBS domain, which has therefore been used in phylogenetic analyses [10,25]. Hence, these major limitations hamper the study of NLRs at various levels such as in the context of plant innate immunity. To address these challenges, in this paper we propose a new LRR motif detection method: LRRpredictor, designed to be more sensitive to motif irregularities than the existing methods like LRRfinder [26] or LRRsearch [27] and to detect irregular and short LRR signatures as are often found in plant NLRs, but not limited to this class. 

We assessed how LRRpredictor behaves within different classes of immune-related receptors that contain LRR domains, such as plant NLRs, RLPs, and RLKs and vertebrate NLRs and TLRs with the aim to provide novel insights into the diversification of LRR domains and their role in the functioning of immune receptors.

## 2. Materials and Methods 

### 2.1. Assembly and Analysis of the LRR Structural Dataset

Various protein domain databases, such as CATH [28], Pfam [29], and Interpro collection [30] were used to obtain a dataset of 611 structure files of proteins annotated to contain LRR domains. These files were processed and filtered out to extract a clean set of LRR chains sharing less than 90% sequence identity using Pisces server [31]. This set containing 178 LRR chains were visually inspected and subjected to LRR repeat delineation based on the distinctive LRR ventral beta-sheet secondary structure pattern. Annotated LRR domains consisting in less than five LRR repeats, as well as incomplete repeats not covering at least five amino acids upstream and downstream of the “LxxLxL” minimal motif were further eliminated. 

Using this procedure, we generated the 90% identity data set, ID90, consisting of 172 N-ter LRR ‘entry’ repeats (N), 1792 LRR ‘core’ repeats (L), and 154 C-ter LRR ‘exit’ repeats (C) (Appendix A). To avoid redundancy in the training data the level of identity has to be further significantly reduced. However, given the small size of ID90 (<180 chains), a trade-off between increase in entropy and loss of data had to be reached. As seen from Figure A1a, a proper inflection point shapes up at around 50% identity and was considered the best compromise in generating a nonredundant set of repeats. In practical terms, the nonredundant ID50 set was generated from ID90 by selecting repeats showing less than eight identical amino acids on a 16 amino acid window centered on the ‘LxxLxL’ minimal LRR motif, i.e., the window comprising five amino acids upstream and downstream ‘LxxLxL’. This nonredundant ID50 set was comprised of 106 N-ter ‘entry’ repeats (N), 659 ‘core’ repeats (L), and 88 C-ter ‘exit’ repeats (C), i.e., ≈40% of the 90ID set (Figure 1, Appendix A). 

Jensen–Shannon divergence (JSD) scores (Figure 1e) were computed using Capra et al. implementation [32], using the BLOSUM62 matrix for background probabilities and a window parameter 0. The phyla distribution shown in Figure 1c was computed using the Environment for Tree Exploration (ETE3) library v3.1.1 [33].

### 2.2. Training and Testing Datasets Construction

In order to provide a representative collection of non-LRR examples, we selected a representative example of each CATH [28] domains’ topology (except LRR) from a nonredundant dataset provided by CATH where all proteins share less than 20% identity or have a less than 60% overlap (cath-dataset-nonredundant-S20 set-09.12.2019). Given potential synchronization problems between various databases used to build the overall learning set comprising (a) the nonredundant 50ID LRRs, containing the ‘entry’-, ‘core’-, ‘exit’-repeats and the flanking nonLRR domains when present and (b) the CATH nonLRR domains—the data was subjected to a third redundancy filter performed with a similar CATH methodology, aimed at eliminating sequences that fail one of the below bounds: the length of the alignment is over 100 and the identity is over 20%.length of the alignment is between 40 and 100 with an identity over 20% and the overlap with respect to both sequences is more than 60%.LRR repeats with alignments lengths ≥16 aa and ≥50% identical (equivalent of at most 8/16 aa constraint imposed initially on the motifs).

The final dataset built as above and used herein for training and testing classifiers, contains 648 LRR core repeats, 100 N-ter entry, and 67 C-ter exit nonredundant repeats (including the LRR domain flanking regions) and 875 non-LRR domains from CATH.

From this set, 1/5th was used to generate the test dataset, while the remaining 4/5 were used to build the training datasets, preserving the class ratio between the sets. The test dataset contains 40,241 amino acid samples of which only 150, i.e., less than 0.4%, are initiating LRR motifs. Similarly, over the training set less than 0.5% of the samples are LRR initiators. The training set was further split into four cross-validation sets that were used for parameter optimization. All these sets are provided in Appendix A.

### 2.3. Feature Selection and Data Pre-Processing

In developing LRRpredictor we tested sequence-based (SeqB) features: solely or combined with structural based (StrB) features. The SeqB features comprise position-specific scoring matrices PSSM over the above discussed 16 amino acids interval summing up to 320 features corresponding to 20 amino acid types over the 16 positions. The StrB features comprise: (a) the three state (H—helix, E—extended, C—coil) secondary structure probabilities, (b) the three class (B—buried, M—medium and E—exposed) residue relative solvent accessibility, RSA probabilities and (c) intrinsic disorder probability—summing up to seven extra structural features per residue, resulting in a total of 432 features per 16 aa window. The structural based predictions were performed with RaptorX-Property software [34,35,36,37]. Sequence PSSMs were computed on Uniprot20 protein sequence database, using HHblits [38,39] that is based on HMM-HMM alignments shown to improve accuracy of alignments at low sequence homology levels. 

In the pre-processing stage, feature variables were normalized, centered, and rescaled, as standard procedure involves. Data whitening using principal component analysis (PCA) decomposition was not used as it did not provide better performance on the tested classifiers.

### 2.4. Machine Learning Model Selection

Several classifiers such as support vector classification (SVC) [40], multi-layer perceptron (MLP) [41,42], and AdaBoost [43] as well as several oversampling techniques such as Adasyn [44] and SMOTE-based varieties [45,46,47], or over- and under-sampling combined approaches SmoteTomek [48] and SmoteEEN [49], were tested and parameter optimized via cross-validation using Scikit-learn library v.0.22.1 [50]. Multiclass estimators for N-entry (N), core (L), and C-exit (C) motif types that use either one-vs.-one or one-vs.-rest approaches were also investigated, but they performed worse than when treating all LRR motifs as a single class.

The best performing classifiers with tuned parameters were further studied in the context of a soft voter (that averages predicted probabilities of the ensemble constituents), and a final predictor, further referred to as LRRpredictor, was chosen based on its out-of-sample performance on test set and overfitting behavior on the training data. LRRpredictor is composed of a set of eight classifiers (C1–C8) that use different strategies and consider all N, L, C motif types as a single class, aggregated within an ensemble based on the soft voting scheme, as shown in Figure 2d.

Classifiers C1–C4 use solely sequence-based features while C5–C8 use both sequence and structural-based features. Classifiers C1 and C5 use the support vector classification (SVC) algorithm [40], with a radial basis function (RBF) kernel, one-vs.-rest (‘ovr’) decision function. The margin penalty and the RBF scale (gamma) parameters were optimized through grid search to 1 and 0.01 for C1 and 1 and 0.001 for C4, respectively. Class imbalance was treated by adjusting the SVM weights inversely proportional to class frequency and class probabilities were inferred using sigmoid probability calibration.

Classifiers C2, C3, C6, C7 use multi-layer perceptron (MLP) [41,42]. A depth of three hidden layers was sufficient to describe the system, as adding additional hidden layers provided little to no difference in out-of-sample performance. The number of hidden nodes for each hidden layer was selected via grid search as follows: C2 (300-250-100), C3 (250-150-100), C5 (250-150-100), C6 (125-100-10). Classifiers C2, C3, C7 use the Limited-Memory BFGS [51] solver, while C6 uses Adam [52] optimizer for stochastic gradient descent [53] with early-stopping over a validation fraction of 0.2. All four classifiers use rectified linear unit (ReLU) activation function [54].

Classifiers C3 and C7 approach the imbalance problem through synthetic resampling using the combined over- and under-sampling method SMOTETomek [48], as implemented in imbalanced-learn library v 0.6.1 [55].

Classifiers C4 and C8 use a ensemble boosting approach—AdaBoost [43]—using tree classifiers of depth 1, as base estimators, SAMME.R real boosting algorithm, and sigmoid probability calibration. A maximum number of 50 base estimators was selected to maximize performance while avoiding overfitting. 

### 2.5. Assembly of Protein Family Sets Containing LRR Domains

In order to investigate LRRpredictor behavior on previously annotated LRR domains from various functional protein groups, we generated a collection of randomly selected 500 representatives from Uniprot50 database (i.e., below 50% identity between themselves at a given minimum overlap—version available at 20.11.2019-release-2019_10) which were annotated by Interpro to contain a LRR domain (IPR032675 and Interpro v77.0 protein2ipr database).

A total of six groups were generated: four groups of sequences of CNLs, TNLs, RLKs, and RLPs protein classes from flowering plants and two groups of TLRs and NLRs from vertebrates. Given the high conservation of vertebrate TLRs this set gathered only ≈350 sequences (Appendix A).

Within the CNL group, there were included only proteins annotated by Interpro to contain a single coiled-coil (CC) domain, a single NBS domain, and a LRR domain in this order, and sequences that contained a different domain organization, such as two annotated NBS domains or a different domain order were not included in the analysis. Similarly, for the TNL group we selected only sequences that contain a TIR-NBS-LRR domain organization. The RLK group was built with sequences displaying a “LRR-TM predicted region-kinase” domain organization, while the RLP group contained sequences with “LRR-TM” organization and did not contain other annotated domains by Interpro. In generating the vertebrate NLR group we included any annotated NACHT or NBS domains followed by a LRR domain annotation without discriminating on the N-terminal domain, as animal NLRs can have upstream of the NACHT/NBS domain a multitude of N-terminal domain types, while vertebrate TLRs group contains sequences with a “LRR-TM-TIR” configuration. Transmembrane predictions were performed using Phobius [56].

In analyzing the length of the LRR domains covered by individual repeat annotations, we used all Interpro annotation codes associated with LRR repeat types. We considered as having the status of ‘annotated as domain‘ LRRs with the IPR032675 label and ‘annotated as repeats’ any amino acids that had attached by at least one predictor part of Interpro collection one of the following tags: leucine-rich repeat (IPR001611), leucine-rich repeat, typical subtype (IPR003591), leucine-rich repeat, cysteine-containing subtype (IPR006553), leucine-rich repeat 2 (IPR013101), leucine-rich repeat 3 (IPR011713), leucine rich repeat 4 (IPR025875), BspA type leucine rich repeat region (IPR026906), CD180 leucine-rich repeat (IPR041281), DUF4458 domain-containing protein, leucine-rich repeat (IPR041403). Annotations referring to the N-ter cap of the LRR domain (IPR000372, IPR041302) were not considered as these are not LRR repeats. 

### 2.6. Assessment of LRR Motif Conservation Across Protein Groups

Intra- and inter-group sequence variability was also analyzed using a subset of 1000 predicted 16 aa extended motifs from each group. In order to avoid a potential bias induced by false ‘entry’ (N) or ‘exit’ (C) repeats, only ‘core’ (L) repeats were used in this analysis. The similarity measure used here is the distance mapping defined by Halperin et al. [57]. This consists of the inner product of BLOSUM scores between each pair of amino acids summed up over the motif span, as this function can be used as a metric distance for several BLOSUM matrices. Considering *d* to be the distance between a pair of amino acids *i* and *j*, that have the *s* (*i*, *j*) BLOSUM score:(1)d(i, j)=s(i,i)+s(j, j)−2·s(i, j)

The distance between two sequences *a* and *b* of equal length *l*, would be the sum of distances of each pair of amino acids *a_i_* and *b_i_* across the length of the sequence:(2)$$D_{a,b}\;=\;\sum_{i=0}^{l}d\left(a_{i},\;b_{i}\right),$$

This definition of distance is expected to reflect amino acids compatibilities, as BLOSUM scores are inferred from amino acid mutation probabilities observed on large datasets. As a BLOSUM matrix we selected an updated version of the original BLOSUM matrix, which was recently recalculated on a large dataset and satisfies the triangle inequality. (RBLOSUM59_14.3) [58,59].

Starting from the above described distance function, we calculated Silhouette coefficients [60] between each pair of groups, and precomputed distances were used for manifold learning using metric multi-dimensional scaling (MDS) [61] as implemented in Scikit-learn library [50]. 

Sequences logos were generated using Weblogo [62], figures showing protein structures were obtained using PyMol [63], while other plots were generated in Microsoft Office or by using Matplotlib library [64].

## 3. Results

### 3.1. Available LRR Domains in Structural Data

A collection of 611 PDB structures previously annotated by several protein domain databases, such as CATH [28], Pfam [29], and Interpro-collection [30] to contain LRR horseshoe architectures was obtained. This collection was used to derive a clean set, ID90, of 178 LRR chains displaying 90% identity that was structurally analyzed in order to structurally delineate the LRR repeats based on the beta-sheet network. By this, a dataset of ≈2100 LRR motifs was obtained, as shown in Figure 1a. It is interesting to note here that less than 20% of these are annotated as LRR motifs in Pfam even though the 178 sequences were derived from known 3D structures. 

The LRR motif annotation of each repeat was performed starting with the first position (L_0_) of the minimal motif ‘L_0_XXL_3_XL_5_’, position that marks the beginning of the ventral side of the horseshoe domain (Figure 1b). Superposition of the 2100 repeats indicates that the structural similarity extends in most of the cases over five positions upstream and downstream of the minimal motif defining a 16 positions region which is referred herein as the ‘extended’ motif (Figure A1d). Due to this, the structural LRR diversity concentrates mainly onto the dorsal side of the horseshoe which imposes onto the curvature and the overall geometry of the domain (Figure 1b). 

As duplications of highly similar LRR repeats within the same LRR domain is abundant in the ID90 set, we opted to perform a second redundancy filter at the level of LRR repeats as described (M&M). This results in the ID50 nonredundant set consisting of ≈850 LRR repeats, that approximates well the ID90 distribution of lengths (Figure A1c), phyla (Figure 1c), and the ratio between marginal N-terminal (N) and C-terminal (C) versus interior motifs (L) (Figure A1b). 

The ‘entry’ N-ter LRR motifs are less regular than the ‘core’ motifs, especially at the first hydrophobic position (L_0_) that is often found solvent exposed, as this position marks the end of the inter-domain linker and the beginning of the LRR domain. By contrast, the ‘exit’ C-ter LRR motifs better resemble the ‘core’ motifs (L) amino acid composition and the conventional LRR motif ‘L_0_xxL_3_xL_5_xx(N/C)_8_xL_10_’ (Figure 1d). Interestingly, the stringency for leucine occurrence sequentially decreases from L_0_ to L_3_ and L_5_ in core repeats, allowing other amino acids to be present in L_3_ and L_5_ more frequently (Figure 1d). This structurally correlates with a larger accessible space of the protein core structure around L_3_ and L_5_ positions, as can be seen from Figure 1b and Figure A1d. It is also worth noting that the third L_-3_ position upstream of LxxLxL has a significant hydrophobic propensity presumably allowing the solenoid to form (Figure 1d). 

Another important facet that has to be carefully pondered is the high phyla bias of the structural data when compared to the baseline phyla distribution of the UniRef50 database. As can be seen from Figure 1c, around 50% of the repeats in ID50 are of mammalian origin while the UniRef50 baseline is of less than 3% in both annotated LRR proteins or any protein. Moreover, the ≈20% plant LRR motifs present in ID50 originate overwhelmingly from RLP and RLK proteins while plant NLRs are poorly represented in this set, with only a single 3D structure recently reported for the ZAR1 NLR protein from *Arabidopsis thaliana* [21,22]. 

### 3.2. Development of the LRRpredictor Method

In order to train a machine learning (ML) estimator for detecting LRR motifs we used an overall dataset comprising the filtered LRR ID50 dataset and a collection of 875 non-LRR domains composed of one representative of each CATH topology (Figure 2a).

As discussed in Section 1, the sequence patterns corresponding to the ≈850 actual *true structural LRR motifs* identified in ID50 are quite common in any protein. We will name here such sequence patterns as *potential motifs*. As expected, Table 1 shows that *potential motifs* occur with more or less equal probability in both the LRR and non-LRR domains of the overall dataset. Moreover, even when taking into account only LRR domains the number of *potential motifs* is larger than the number of *true structural LRR motifs* (Table 1). This allows the ML estimators to learn to detect *true motifs* from the far larger set of *potential motifs* by taking into account the larger 16 amino acid sequence context in which the *true motifs* are embedded. In this way the method developed herein can be used not only to delineate repeats in a given LRR domain but also to discriminate between protein products that do not have LRR domains from those hosting such domains.

In developing LRRpredictor we tested ‘sequence-based’ features based on position-specific scoring matrices-PSSMs either solely or combined with ‘structural-based’ features as described in Section 2 (Figure 2c). PSSM profiles are expected to provide context information on the overall sequence, to highlight the key amino acids position that are conserved, as the amino acids scores are derived from amino acid substitution probabilities conditioned by the homologues family they belong to. Therefore, it is expected that irregular LRR motifs would be more detectable when using sequence profiles, rather than amino acid sequence alone. 

The dataset was split into five parts: one part was initially separated as test set and the other four were used as a training set in parameter tuning using a four-fold cross-validation (CV) approach, where models were iteratively trained on three of the CV sets and tested on the remaining fourth (Figure 2b). A pool of estimators (representing algorithms for classification) that used either (1) sequence-based or (2) both sequence and structural features and (3) various imbalance class treatments were optimized via cross-validation. Finally, best performing estimators were studied in the context of an ensemble estimator. The selected ensemble classifier, further referred to as LRRpredictor is a soft voter aggregating eight classifiers C1–C8 (Figure 2d) which were trained to detect the LRR motif starting position—i.e., L_0_ position from the minimalistic LRR motif ‘L_0_xxL_3_xL_5_’.

Finally, LRRpredictor was trained on the entire training set (all four CV sets) and tested on the test set which had been set aside.

### 3.3. Assessment of LRRpredictor Performance

The precision of LRRpredictor given by the fraction of true-positives (TP) predicted results over the sum of true-positives (TP) and false-positives (FP) varies between 89% and 97% on the test set and within cross-validation sets (Figure 3a). Similarly, the recall (also known as sensitivity), given by the fraction of TP over TP + false-negatives (FN) varies between 85% and 93%, while the F1-score (representing the harmonic mean between precision and recall) varies between 87% and 95% on the test set and cross-validation sets (Figure 3a,b).

### 3.4. LRRpredictor Behavior on Protein Families Containing LRR Domains

As the available structural data is scarce, we further evaluated the extrapolation capabilities of LRRpredictor on a set of LRR domains annotated in Interpro collection. Groups of the most representative protein functional classes containing LRR domains were generated as follows: four groups from flowering plants—resistance proteins (CNL_plants_ and TNL_plants_) and extracellular receptors (RLK_plants_ and RLP_plants_) and two groups from vertebrates—NLR_vert_ and TLR_vert_ as described in Section 2.

Selected sequences from each group were subjected to LRRpredictor motif detection. The repeat length distribution of the predicted LRR repeats (Figure 4a), is consistent with previously reported lengths within all protein groups of the seven type Kobe–Kajava (KK) classification [14,65]. The repeat length distribution of extracellular LRR domains (RLK_plants_, RLP_plants_, and TLR_vert_) show a sharp peak at 24 amino acids, in agreement with the most frequent repeat length within plant-specific (PS) from KK classification [14,65]. As they often contain large helices over the dorsal side of the LRR horseshoe, vertebrate NLRs repeats have longer lengths (25–30 aa) as previously shown by the same classification, while plant NLRs (CNL_plants_ and TNL_plants_ ) have a larger distribution with a lower peak shaping up toward lower value side (20–24 aa) of repeat lengths range.

As seen in Figure 4a, repeat lengths are rarely found outside the 19–35aa range, cases in which prediction becomes ambiguous. Too short repeats are improbable due to structural constraints and might indicate false positive predictions. Similarly, too long repeats—over 40 amino acids—could indicate either the presence of undetected repeats (false negatives) or cases in which an insertion or ‘island’ shapes up protruding the horseshoe structure (Figure 4a). Very large gaps between LRR motifs (more than 100 aa) were not included in computing the length distribution as these are rather indicating the presence of an inserted domain flanked by two LRR domains.

We further analyzed the percent of the annotated LRR domain span that is covered by LRRpredictor and compared the predicted LRR motifs to LRRfinder [26] and LRRsearch [27] predictions and to the existing motif annotations from Interpro collection. In doing so we defined as predicted repeats motifs separated by 15–35 amino acids. Predicted motifs that superpose or cluster within 15 amino acids were counted only once, while when the distance between two motifs was higher than 35, the repeat was considered to be a potential terminal repeat or contain a domain break and the first 24 aa of such a stretch was assigned as a predicted repeat, given that this is the most frequent repeat length over the structural data. 

The repeat coverage of analyzed LRR domains predicted by LRRpredictor, LRRfinder and LRRsearch were compared to the extent Interpro repeat annotations using a coverage percentage (CP) defined as the ratio between the sum of predicted/annotated repeat length vs the overall LRR domain length (Figure 4b). 

Plant NLRs from flowering plants show the lowest level of repeat annotation as 75% and 50% of CNL and TNL LRRs in Interpro lack *any* repeat annotation resulting in CP = 0% (Figure 4b). In comparison, repeats in vertebrate NLRs are better annotated in a CP ranging within 20–60% of the LRR domain. Even higher Interpro repeat annotations are shown by the extracellular plant and vertebrate receptors with CP ranging most frequently between 30% and 80% of the LRR domain size (Figure 4b). For all six receptor classes analyzed herein, both LRRfinder and LRRsearch slightly increase the LRR coverage as compared to Interpro annotations especially in the case of extracellular receptors, with LRRsearch surpassing LRRfinder in the case of plant NLRs (Figure 4b).

As also can be seen from Figure 4b, in comparison to Interpro and the two predictors mentioned above, LRRpredictor covers far larger regions of LRR domains with coverage percentages (CP) exceeding 60% and almost complete coverage in over 50% in all six groups (Figure 4b). It is interesting to note that Interpro annotation of extracellular LRR domains also include the N-terminal cap region, that is *not* formally a LRR repeat. This results in the fact that LRRpredictor covers in most cases only ≈90% of this domain, instead of 100% as in NLR groups (Figure 4b).

### 3.5. Predicted Repeats Consensus in Each Class 

Further, the amino acid composition of the predicted LRR motifs was investigated solely on the ‘core’ predicted LRR repeats, i.e., repeats that are flanked by other predicted repeats within a 15–35 aa range. In short, the results presented below clearly indicate that LRRpredictor is able to detect and reproduce all the consensus motifs previously defined for well-studied classes of RLKs, NLRs, and TLRs (Figure 5). 

This is especially the case for vertebrate NLRs. The consensus follows the ribosomal inhibitor (RI) type - ‘x_-3_xxL_0_xxL_3_xL_5_xx(N/C)_8_xL_10_xxxgoxxLxxoLxx’ [14,65], with position ‘-3’ being less relevant for this class of repeats. Additionally, the vertebrate TLRs predicted motif consensus matches the 

“T” type motif: L_-3_xxL_0_xxL_3_xL_5_xxN_8_xL_10_xxL_13_xxxx(F/L)_18_xxL_21_xx 

defined in Matushima et al. classification [13] rather than the less encountered 

“S” type motif: L_-3_xxL_0_xxL_3_xL_5_xxN_8_xL_10_xxL_13_Px(x)LPxx.

In the case of plant extracellular receptors, the predicted motifs from RLK_plant_ and RLP_plant_ groups show a prolonged pattern that is in perfect agreement with the plant-specific (PS) type from Kobe and Kajava classification [14,65]—L_-3_xxL_0_xxL_3_xL_5_xxN_8_xL_10_(S/T)_11_GxIPxxLxxLGx. Interestingly, the kinase containing receptors (RLK) have a more prominent consensus (Figure 5). 

On the other hand, the predicted motifs in plant NLRs comprising the CNL_plant_ and TNL_plant_ groups display a remote similarity with the cysteine-containing (CC) type as defined by Kobe and Kajava classification [14,65]:

“(C/L)_-3_xxL_0_xxL_3_xL_5_xxC_8_xxITDxxOxxL(A/G) xx”—where O is any nonpolar residue. 

While the extended motif is satisfied (16 aa), a difference worth noting is that in both CNL and TNL groups cysteine is rare in position ‘-3’ and outside this region any similarity with CC-type ends. Both plant NLR groups mainly confine their consensus to only the minimal L_0_xxL_3_xL_5_ motif, with TNL extending it a little bit with C_8_ position. By contrast, in plant extracellular receptors the consensus expands beyond the 16 amino acids of the ‘extended’ region covering all the four sides of the LRR solenoid. Despite being analogous in composition, the TNL_plant_ group consensus is more pronounced, especially at positions C_8_ and L_11_ (Figure 5).

In all six classes L_0_, L_3_, and L_5_ of minimal motif are as expected overwhelmingly hydrophobic, with all three positions occupied by leucine in around 50% of the cases, except CNLs where leucine occurrence seems less stringent (Figure 6). When compared to the Kobe and Kajava classification, the majority of the motifs fall under the expected class and very few cross terms are seen between them (Figure 6). 

CNLs and TNLs seem more dispersed even on a shorter 11 amino acid window consensus (W11), while the extracellular receptors obey in over 60% of the cases the corresponding W11 pattern that is shared simultaneous by all three classes (Figure 6).

### 3.6. LRR Motifs Variability Across Classes 

Sequence variability is of critical importance for LRR domain function and in contrast to their common structural pattern, a wide spread in the sequence space is expected. To assess this, we analyzed the extended motifs, predicted by LRRpredictor, both the intra- and inter- group sequence similarity. This was performed over subsets of randomly selected 1000 examples of ‘core’ (L) motifs from each group. We selected as similarity measure a metric distance function [57] derived from BLOSUM scores which reflect the structural compatibility between amino acids, as described in Section 2. Using this metric, we calculated the distance between each predicted LRR motif from all groups and analyzed how these distances behave intra- and inter-groups. 

Intra-group all-vs.-all distances distribution shows that the extracellular groups RLK, RLP from plants and TLRs from vertebrates form a denser group in terms of conservation, than plant and vertebrate NLRs (Figure 7a left). Figure 7b shows the silhouette coefficients. These scores show how separated two given clusters are, based on the distance between samples from each group, the maximal value of 1 corresponding to perfectly separated clusters, value 0 corresponds to clusters that coincide, while negative values with a minimum of -1 correspond to the case where samples from one group actually cluster better with the opposite group that is being compared. Silhouette coefficient of all versus all analyzed groups indicate that the NLR groups form a rather overlapping cluster, that has an increased variability among its sample motifs (i.e., expanded cluster) (Figure 7b left). Extracellular plant receptors RLK and RLP clusters are overlapping and have a more reduced span in terms of variability (i.e., more conserved motifs), while vertebrate TLR overlap plant RLK and RLP receptors have a slightly increased variability (Figure 7a,b left). Interestingly, within the minimal LRR motif region ‘L_0_XXL_3_XL_5_’ there are no significant differences between groups (Figure 7a,b right).

To have an overall view on the sequence dispersion in each protein class containing LRR domains Figure 7c shows the 2D embedding of the high dimensional sequence space of both the extended and minimal motifs of each class. Nonetheless such a reduction gives only a rough representation of distance relations between clusters in the original space as the normalized stress parameter (*stress*-1) of this 2D embedding is 0.25 and 0.21 for the extended and minimal motif space, respectively [66].

### 3.7. LRRpredictor Specificity Tested on Solenoid Architectures 

From a structural point of view the LRR protein architecture belongs to the larger class of solenoidal architectures which are defined by specific repeated structural patterns. Given the repetitiveness of such structures we asked if LRRpredictor is able to discriminate between LRR motifs and other repetitive sequence patterns. The main candidates considered for possible misclassifications are two classes of beta sheet repeat proteins—which are the closest structural relatives of LRR domains: pectate lyases (PeLs) and trimeric LpxA architectures and two helical repetitive classes: armadillo and ankiryin architecture (Figure A2b). To this end, 50 sequences from each of the above four classes annotated as such by Interpro were randomly selected from UniRef50. Figure A2a shows the probabilities returned by LRRpredictor that the potential motifs occurring in the 200 sequences are true LRR structural motifs. As can be seen in all four classes taken into account, the vast majority of potential motifs have a probability lower than 10% to be true motifs. Only 0.1% of such sites show a probability between 10% and 20% to be true motifs and none of these sites reaches a threshold of 40% for being a true LRR motif (Figure A2a). From a technical point of view this result shows that LRRpredictor is highly specific for LRR domains. On the other hand this result is even more interesting from a structural and biological point of view indicating that even if LRRs and PeLs were considered to be members of the same LRR superfamily [67] the structural principles upon which they are built are different and presumably the two classes have diverged very early in evolution.

## 4. Discussion

Given the high number of indeterminacies generated by sequence variability, a proper annotation of LRR motifs and the correct delineation of repeats is critical in identifying potential protein–protein interaction sites of LRR domains. 

Here, we show that LRRpredictor is able to address this problem and by this, can be of use as a new tool in the analysis of especially plant NLR sequences that display a larger variability and irregularity as compared to other LRR domains [9,68]. This often results in the superposition or presence in less than a minimal repeat distance of potential alternative LRR motifs, as can be seen from Figure 8 illustrating such indeterminacies found on a 150 amino acid stretch from the potato CNL Gpa2 LRR domain.

Given the scarcity of structural learning data consisting of less than 180 LRR structures with lower than 90% identity and only ≈850 motifs at hand in ID50 (<50% identity), in order to maximize LRRpredictor extrapolation abilities, the method was set to rely on aggregating a collection of eight classifiers based on different strategies, two of them designed to perform a massive oversampling of the real data (Figure 2).

In this context, LRRpredictor shows to perform well, with overall *precision*, *recall*, and *F1* scores ranging between 85% and 97% on both test and cross validation sets (Figure 3a). In addition, LRRpredictor increases its performances when taking into account only the ‘core’ repeats (L), as the main prediction problems relate only to the N-‘entry’ repeats (N)—i.e., the first repeat of the LRR domain (Figure 3a, Table A1). This can be explained in part by the increased irregularity of the sequence in this region, but also by the small sample size of the N-‘entry’ (N) motifs when compared to the ‘core’ (L) motifs.

It is also important to note here the fact that false positives are almost never found in nonLRR domains but always in proteins containing LRR domains (Table A1). Here, such false positives shape up in close vicinity to the marginal repeats—where the LRR motif characteristics are more diffuse, or in linkers or different domains neighboring the LRR, but found in a ‘one repeat range’ to the N-entry motif. 

Other false predictions are caused by alignment artefacts. These yield to an offset of 1–3 amino acids in the predicted LRR motif starting position. Alignment artefacts are also frequently seen in regions with high beta structure propensity of insertion loops or ‘islands’ protruding from the LRR domain structures. This is mainly due to the fact that the multiple alignment on which PSSM relies forces the protruding loop in the queried sequence to align to regular repeats in the template LRRs of the database. 

Unfortunately, the number of such insertion loops or ‘islands’ is so small in ID90/ID50 that estimators cannot learn from the existing data to discriminate such false positives. Thus, only careful structural analysis performed in later modelling stages can handle such cases.

Results on both cross-validation and test sets show that estimators using structural features in addition to the sequence based features (C5–C8) perform on average only slightly better compared to those sequence based only (C1–C4), with some interesting improvements on F1 scores (Figure 3b). This only marginal improvement may indicate that RaptorX-Property [34] training on the overall structural database that might have marginally overlapped with our testing dataset did not affect the results. Nevertheless, C5–C8 are expected to be better extrapolators (Table A1), while the structural predictions on which (C5–C8) are based, and that are present in the output file can prove instrumental in further dealing with ambiguous cases where two LRR motif signatures partially superpose or are within the limit of a repeat.

Figure 3c and Figure 4b compare predictions of three existing engines. LRRpredictor outperforms LRRsearch [27] and LRRfinder [26]. This is expected as the two previous methods were designed to focus mainly on specific LRR classes such as vertebrate TLRs or NLRs, respectively, while LRRpredictor relies on a newer larger dataset and was designed to identify LRR motifs in general. However, despite focusing on specific protein classes, both LRRsearch and LRRfinder show comparable efficiency in covering annotated LRR domains in plant extracellular receptors but decreased capabilities on plant NLRs (CNLs and TNLs) (Figure 4b). Furthermore, both LRRsearch and LRRfinder were intended for fast computation and they use a predefined PSSM matrix computed on a curated collection of LRR domains, instead of performing case by case basis sequence profiles as our method does.

However, the increased performance of LRRpredictor comes with an attached computational cost and is not easily scalable for scanning large protein sequences databases such UniprotKB. The main reason for this is that generating case by case sequence profiles and performing predictions for each estimator aggregated in LRRpredictor is more computationally demanding than LRRfinder and LRRsearch workflow.

Another matter of concern was related to the phyla bias of the database on which LRRpredictor relies, as ≈50% of ID90 have mammalian origin while the share of mammalian—from total—annotated LRRs in UniRef50 is only 2% (Figure 1c). Moreover, groups such as plant NLRs are extremely poorly represented, as only very recently the first plant NLR structure was reported [21,22]. 

In this context, in order to investigate the extrapolation capabilities of LRRpredictor we used a set of LRR domains annotated in Interpro collection from the six most representative immune receptor classes: R-proteins and extracellular receptors from flowering plants (CNL_plants_, TNL_plants_, RLK_plants_, RLP_plants_) and their vertebrate counterparts (NLR_vert_, TLR_vert_). The LRR motifs predicted by LRRpredictor show a good coverage of the LRR domains annotated by Interpro and follow the expected repeat length distribution for all these six classes [14,65] (Figure 4a,b). Moreover, the predicted motifs reproduce the expected LRR motif consensus of each protein class (Figure 5) from Kobe and Kajava classification [14,65]. Combined, these indicate that LRRpredictor is able to extrapolate well in different LRR motif classes which is especially important for plant NLRs.

Analysis of LRRpredictor detected motifs showed clear differences between the six classes within the extended 16 aa motif. Whether variation in these extended motifs directly relate to the functional diversification of the different receptor classes still remains to be addressed. By contrast, within the minimal 6 aa LRR motif region—L_0_XXL_3_XL_5_—there are no significant differences between the six groups (Figure 7). This might suggest a common root of minimal structural criteria imposed by the solenoidal architecture from which the six classes have diverged to fulfil specific tasks in specific environments. For receptor function, such a solenoidal domain organization in which only three positions over a ≈25 repeat length are loosely conserved has two-fold evolutionary advantages: first the solenoid architecture ensures a large solvent exposed surface area [10] and second a high sequence variability can be achieved without disturbing the tertiary structure.

The increased conservation seen at the level of the extended motif among all three extracellular LRR classes—plant RLK, RLP and vertebrate TLRs—when compared to plant and vertebrate NLRs could be related to N-glycosylation and the constraints imposed by the extracellular environment. On the one hand, plant NLRs recognize directly or indirectly a suite of pathogen effectors or (perturbations) of their host targets conferring host specific immunity. Single amino acid changes in the effector can already be detected or are sufficient to evade recognition by a NLR, resulting in a co-evolutionary arms race between pathogen effectors and host immune receptor [69,70]. In contrast, extracellular LRRs recognize often conserved microbial patterns to confer basal immunity thus lacking such a strong driver for diversification [71]. Vertebrate NLRs act more like basal immune receptors in innate immunity, recognizing conserved microbe-associated molecular patterns (MAMPs). The greater diversifying selection imposed by fast-evolving effectors may therefore account for the co-evolution of structurally highly variable LRR motifs in plant NLRs. In the future, it will be interesting to relate our LRR structural annotations to specific functional NLR sub-classes. This is relevant, for instance, as some plant NLRs-types are described to have a downstream ‘helper’ function rather than a role as a canonical ‘sensor’. 

Another aspect is that LRR domains in plant NLRs have a dual role. They not only contribute to pathogen recognition, but also negatively regulate the switch function [72]. Hence, it will be interesting to link LRR structural annotations to specific intramolecular domain interactions between LRRs and other NLR subdomains to better understand the co-evolution of protein domains in NLRs. It is shown that subtle mutations in the interface between LRR and NB-ARC can have a major effect on NLR functioning, often resulting in constitutive immune activation or a complete loss-of-function [17,72]. This shows the tight link between structural and functional constraints underlying the shaping of NLRs in plants. Additionally, the link between LRR structural annotations and complex formation with other host proteins will be interesting to assess. LRR domains are known to interact with other components like chaperones (e.g., SGT1) which are required for proper NLR folding and functioning [73], or kinases (e.g., ZED1, RKS1) [21,22,74] but also NLR hetero- and homodimers are often formed [9,75,76] which could impose additional structural constraints on the shape and irregularity of LRR domains in plant NLRs. 

## 5. Conclusions

The results presented herein indicate that LRRpredictor shows a good performance on the available 3D data and good extrapolation capabilities on plant NLRs (CNL/TNL), which are poorly represented in the training dataset. Predicted LRR repeats using LRRpredictor significantly increase the coverage of Interpro annotated LRR domains from main immune receptors groups. In addition, these predicted repeats are consistent with previously defined motif consensuses from all studied groups and also follow the repeat length range specific to each class. In conclusion, LRRpredictor is a tool worth using in research topics related to understanding immune receptors functions and structure-informed strategies for pathogen control technologies.

## Figures and Tables

**Figure 1 genes-11-00286-f001:**
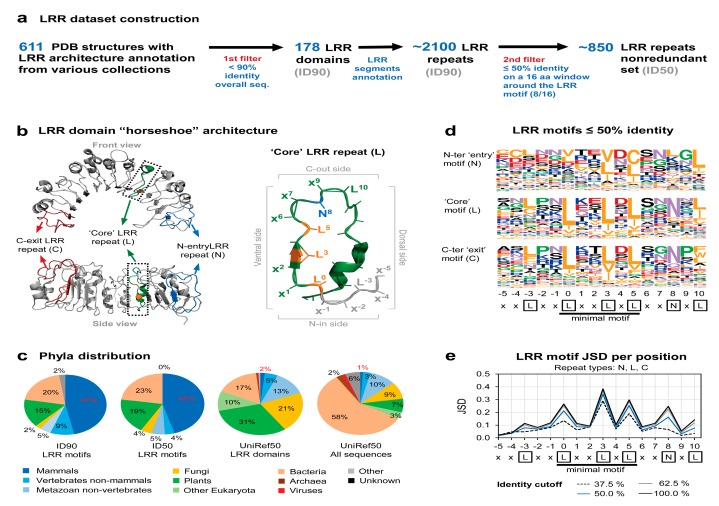
Available leucine-rich-repeat (LRR) domains in structural data. (**a**) LRR structural dataset construction. (**b**) LRR domain horseshoe architecture illustrated on the only plant NLR cryo-EM structure available—ZAR1—from *Arabidopsis thaliana* (left) and zoom-in view of a LRR repeat (right) (PDB: 6J5W). The hydrophobic positions in the minimal ‘L_0_xxL_3_xL_5_’ motif are shown in orange. The first N-entry repeat (blue) and the last C-exit repeat (red) are also mapped on the structure. (**c**) Phyla distribution of the initial LRR motif set ID90, the 50% identity trimmed LRR motifs set (ID50), annotated LRR proteins and all proteins from the UniRef50 database (from left to right). Percent values corresponding to the mammals group are shown in red. (**d**) Frequency plot of amino acid composition of the N-entry, core and C-exit motifs on the 50% identity trimmed set. Amino acids are colored according to their properties as follows: hydrophobic (yellow), acidic (red), basic (blue), asparagine and glutamine (purple), proline and glycine (green), others (black). (**e**) Jensen–Shannon divergence (JSD) score for each position of the LRR motif at different identity thresholds. Higher values show increased conservation.

**Figure 2 genes-11-00286-f002:**
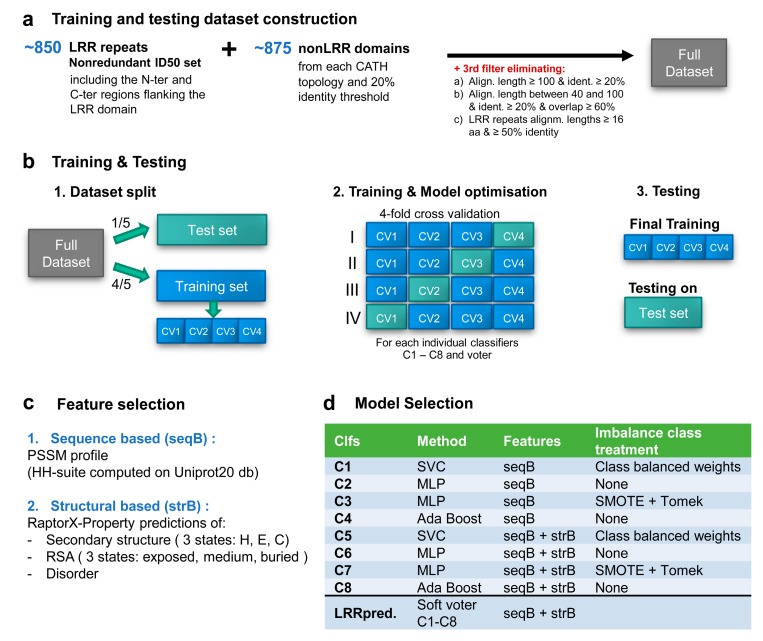
LRRpredictor training and testing workflow: (**a**) training and testing dataset construction. (**b**) schematic representation of the training and testing procedure, (**c**) selected features, and (**d**) selected classifiers aggregated into LRRpredictor.

**Figure 3 genes-11-00286-f003:**
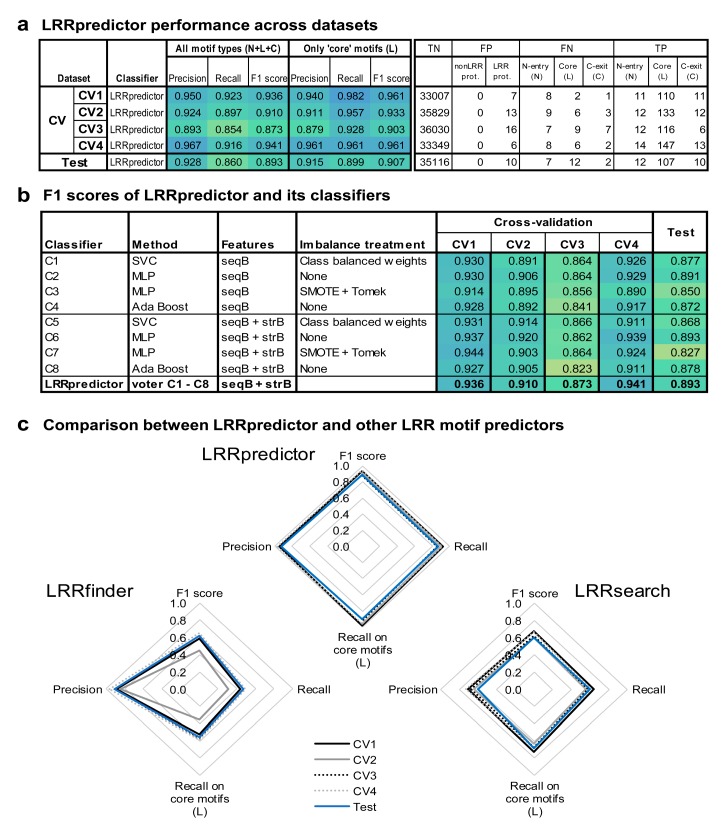
LRRpredictor performance analysis: (**a**) LRRpredictor performance across datasets: precision, recall, and F1 scores are shown either considering all the LRR motif types (N-entry, core, and C-exit types), either solely core motifs (L); also shown are the true negative (TN), false positive (FP), false negative (FN) and true positive (TP) counts. (**b**) F1 scores of LRRpredictor and its individual classifiers. (**c**) Comparison between LRRpredictor and other LRR motif predictors: LRRfinder [26] and LRRsearch [27] (computed on their webservers using default parameters).

**Figure 4 genes-11-00286-f004:**
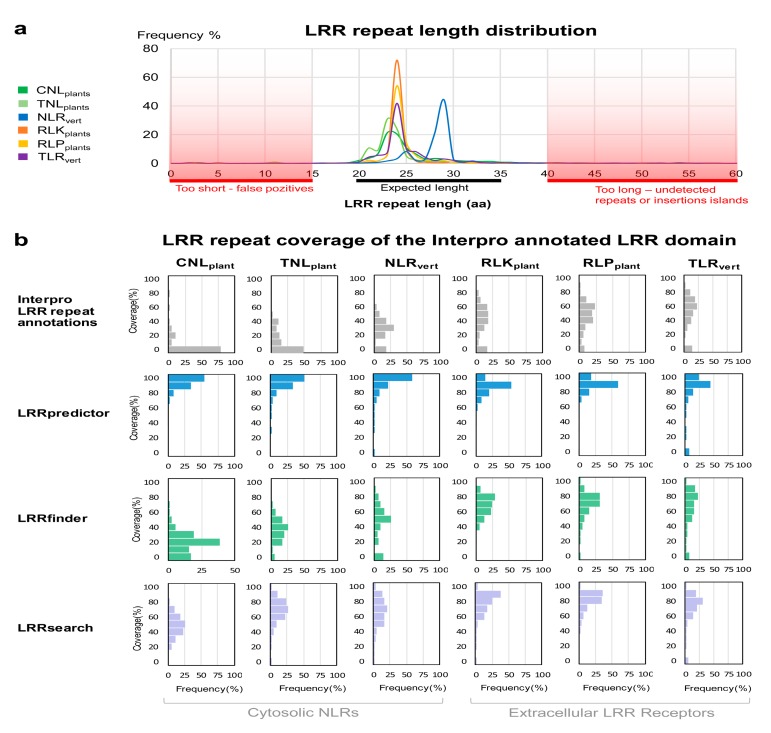
LRRpredictor behavior on Interpro annotated LRR domains from different classes. (**a**) Length distribution of the predicted repeats using LRRpredictor within each protein group. C-terminal motifs were not used in computing the distribution. Repeat lengths size prone to ambiguity—i.e., either too short (potential FP) or too long (potential FN)—are shaded in red. (**b**) Distributions of the Interpro annotated LRR domain length that is covered by Interpro LRR repeat annotations (grey) or by predicted repeats using LRRpredictor (blue), LRRfinder (green), and LRRsearch (purple). Coverage percent distributions are shown within each protein group.

**Figure 5 genes-11-00286-f005:**
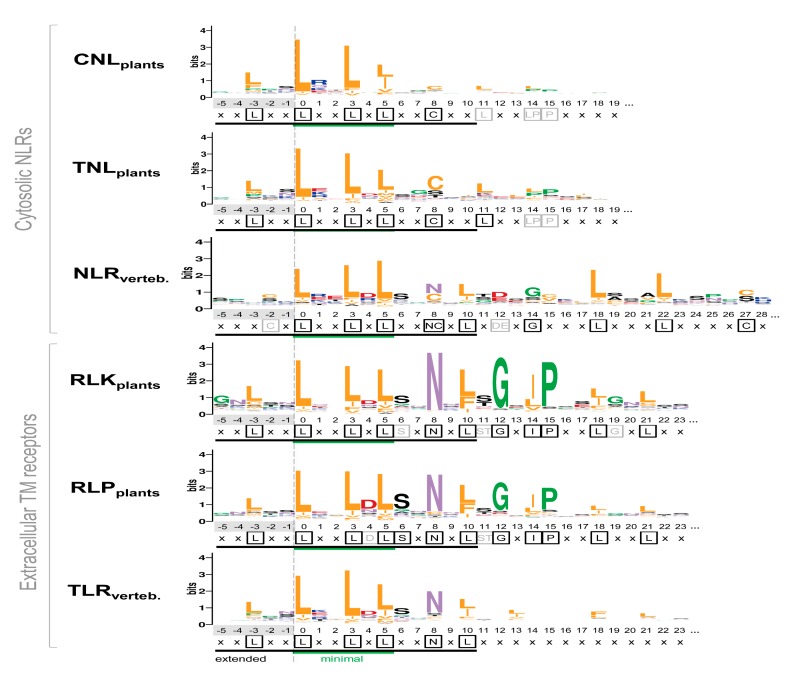
Consensuses of the LRR motifs predicted by LRRpredictor across different classes. Logo heights correspond to amino acid relative entropy (in bits), higher heights implying higher conservation. A consensus for each class is displayed bellow each logo, highly conserved positions being shown in black boxes, while less conserved in gray. Minimal motif ‘L_0_xxL_3_xL_5_’ (green line) and the extended motif (black line) are indicated below each logo. Amino acids are colored according to their properties as in Figure 1d.

**Figure 6 genes-11-00286-f006:**
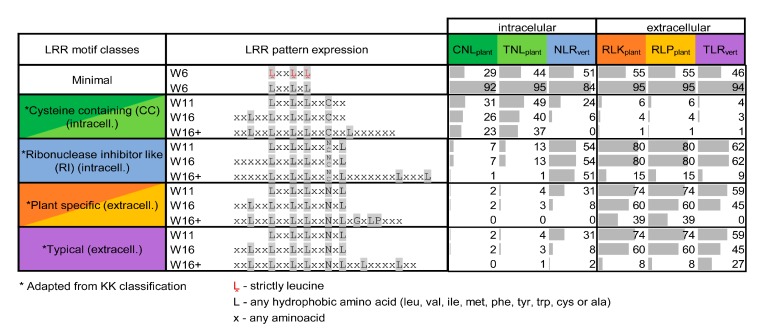
Distribution of LRR motif types defined by Kobe and Kajava (KK) [14] predicted with LRRpredictor across the six receptor classes. As the motif consensuses from KK classification were very strict, we adapted these consensuses to different sequences windows (W6, W11, W16, or more) centered around the minimal motif as shown in the table. Percentages of the predicted motifs compatible with each consensus are shown with grey bars.

**Figure 7 genes-11-00286-f007:**
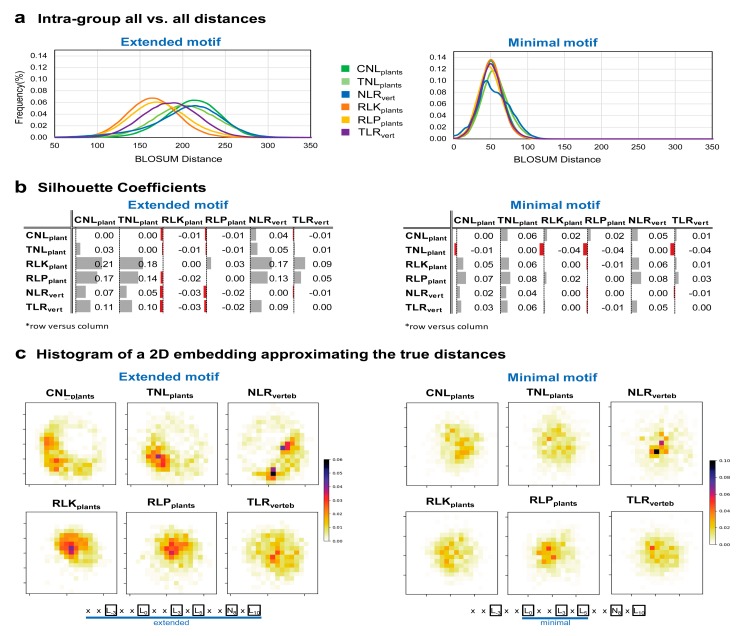
LRR motifs variability in different protein classes. (**a**) Intra-group all-vs.-all distances on the extended (left) and minimal (right) motif (**b**) Silhouette coefficients inter-groups extended (left) and minimal (right) motif. (**c**) Histogram of a 2D embedding approximating the true distances between points for the extended (left) and minimal (right) motif. Histograms were computed using a 20 × 20 bins grid. Extended and minimal motif histograms cannot be compared as they refer to different sequence spaces.

**Figure 8 genes-11-00286-f008:**
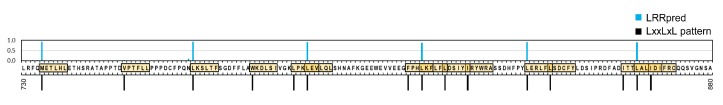
LRR motif and repeat indeterminacies onto a 150 aa stretch in Gpa2 potato NLR. Potential motifs that follow the minimal ‘LxxLxL’ pattern (where L is any hydrophobic amino acid) are illustrated above the sequence with black bars and yellow highlight, while LRRpredictor results are shown above with blue bars.

**Table 1 genes-11-00286-t001:** Occurrence of LRR sequence patterns in the overall dataset used to train the machine learning (ML) estimators.

			Full Training & Testing Dataset (CV 1-4 and Test Sets)			
			NonLRR Proteins		LRR Proteins	
LRR-Like Pattern		Total Number	False Motifs	True Motifs	False Motifs	True Motifs
ḼxxḼxḼ	(3Ḽ)	296	114	0	27	155
ḼxxḼxŁ/ḼxxŁxḼ/ŁxxḼxḼ	(2Ḽ)	1,060	773	0	147	140
ḼxxŁxŁ/ŁxxŁxŁ/ŁxxŁxḼ	(1Ḽ)	7,239	5,875	0	1,192	172
LxxLxL		12,247	10,149	0	1,417	681
LxxLxLxxN		811	438	0	76	297
LxxLxLxxC		273	163	0	41	69
LxxLxLxx(N/C)xL		618	269	0	39	310
Number of predicted positions (16 aa sliding windows):			148,540		25,658	
						**815 LRR motifs**

Ḽ—strictly leucine; Ł—hydrophobic without leucine (I, V, M, F, W, Y, C, A); L—hydrophobic (L, I, V, M, F, W, Y, C, A); x—any amino acid.

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
