# Peer review of "LRRpredictor—A New LRR Motif Detection Method for Irregular Motifs of Plant NLR Proteins Using an Ensemble of Classifiers"

_genes, 2020, doi:10.3390/genes11030286_

Round 1
Reviewer 1 Report
The paper describes a much needed improved way to predict LRRs. The major improvement over the previous methods comes from training the algorithm on a wider set of proteins, including plant NLRs that have more flexible LRR patterns. The software is timely as the number of plant genome data, and NLR data specifically grows. The software is robustly tested against other similar programs and shows superior performance (however as the authors correctly point it out this could be specific to the dataset used in this study to train LRRpredictor). Figure 4 is especially outstanding as it shows that previous algorithm do not cover most of the LRR of the cytoplasmic plant NLRs.
I have a few major comments:
- It is not possible for me to evaluate the software without seeing the actual program and testing it. Please, make available for review (password protected is ok).
- The algorithm performs best on core LRR motifs, however its performance still suffers on N-terminal and C-terminal motifs. From materials and methods it is not clear if separate classifier models have been built for the 3 LRR categories and if the parameters were optimized for each class separately.
- Since the program performs best on core LRRs, it might be worthwhile exploring an iterative prediction approach, predicting the core motifs first, and then scanning the ends for N and C terminal motifs that should be in correct 'register' with the core motifs.
Minor points
line 93 "representant" needs to be removed or reworded.
lines 94-95. Please, reword the internal clause
line 167. "low homology" should be replaced with 'low sequence similarity' or clarified that you mean structural homology, and not sequence homology.
line 286. please, reword the first clause to simplify and make it more clear.
line 368 "predictor prediction behaviour" seems redundant, it might be best to reword this section to summarize the key point of the section, or re-word to "LRRpredictor performance on"
line 491-492. One sentence paragraph. please, expand or merge with previous one.
Author Response
Please see the attachement

Reviewer 2 Report
The authors present the results of training a statistical machine learning method for identifying leucine-rich-repeat (LRR) protein sequences, specifically targeting plant proteins containing these repeats. Overall, the methodology is sound, the results are well-presented, and the work is of very high quality.
The one minor concern I have is with potential false-positives. The authors indicate that, due to their short length and repetitive nature, LRR-like motifs are expected to occur 'randomly' in many structural contexts, which makes LRR detection potentially difficult. Although the authors employ an impressive training strategy that includes non-LRR structures and appropriate use of cross-validation and out-of-sample testing, the extent to which LRR-like motifs were included in the negative training/testing data sets was not clear from the manuscript. If LRR-like motifs were largely absent from the negative training/testing data, this could lead to inflated accuracy estimates, as the particular cases expected to be the most difficult to differentiate (true LRRs vs LRR-like sequences) would be excluded from training and/or testing.
If the authors could provide some analysis of their method's performance in these challenging negative cases, it would strongly improve the impact of the manuscript.
